# A Plasmid-Based Fluorescence Reporter System for Monitoring Oxidative Damage in *E. coli*

**DOI:** 10.3390/s22176334

**Published:** 2022-08-23

**Authors:** Hariharan Dandapani, Pasi Kankaanpää, Patrik R. Jones, Pauli Kallio

**Affiliations:** 1Molecular Plant Biology, Department of Life Technologies, University of Turku, FI-20014 Turku, Finland; 2Turku BioImaging and Turku Bioscience Centre, University of Turku, FI-20014 Turku, Finland; 3Turku BioImaging and Turku Bioscience Centre, Åbo Akademi University, FI-20500 Turku, Finland; 4Department of Life Sciences, Faculty of Natural Sciences, Imperial College London, London SW7 2BX, UK

**Keywords:** stress response, oxidative stress damage sensor, iron–sulphur cluster, reactive oxygen species, lowered growth temperature, plasmid-based fluorescent reporter, transcriptional regulation, *E. coli*

## Abstract

Quantitating intracellular oxidative damage caused by reactive oxygen species (ROS) is of interest in many fields of biological research. The current systems primarily rely on supplemented oxygen-sensitive substrates that penetrate the target cells, and react with ROS to produce signals that can be monitored with spectroscopic or imaging techniques. The objective here was to design a new non-invasive analytical strategy for measuring ROS-induced damage inside living cells by taking advantage of the native redox sensor system of *E. coli*. The developed plasmid-based sensor relies on an oxygen-sensitive transcriptional repressor IscR that controls the expression of a fluorescent marker in vivo. The system was shown to quantitatively respond to oxidative stress induced by supplemented H_2_O_2_ and lowered cultivation temperatures. Comparative analysis with fluorescence microscopy further demonstrated that the specificity of the reporter system was equivalent to the commercial chemical probe (CellROX). The strategy introduced here is not dependent on chemical probes, but instead uses a fluorescent expression system to detect enzyme-level oxidative damage in microbial cells. This provides a cheap and simple means for analysing enzyme-level oxidative damage in a biological context in *E. coli*.

## 1. Introduction

Aerobic life on Earth depends on molecular oxygen for maintaining cellular respiration and the assembly of oxygen-containing biomolecules and inorganic compounds. Although O_2_ is essential for various biological processes, it is also a significant stress factor for living systems, as it is linked to the formation of reactive oxygen species (ROS) that may readily harm vital metabolic functions [1]. ROS include oxidising agents, such as superoxide (^•^O_2_^−^), and derived species, such as hydrogen peroxide (H_2_O_2_) and hydroxyl radical (^•^OH), which inflict direct and indirect oxidative damage by reacting with lipids, proteins, and nucleic acids in the cell [2,3,4,5]. One specific group of cellular components that are particularly vulnerable to ROS-induced oxidative damage is the enzyme-bound iron–sulphur (Fe-S) cluster cofactors. These organometallic assemblies of iron and sulphide are required for the catalytic activity of many proteins involved in redox reactions, and are essential for a wide range of central biological processes in nature, including respiration, photosynthesis, and nitrogen fixation [6].

*E. coli* has several functionally interlinked response mechanisms dedicated to the biogenesis and upkeep of Fe-S clusters to ensure the integrity of Fe-S proteins under different metabolic conditions, as reviewed in [7,8,9]. The primary system called ISC (*i*ron–*s*ulphur *c*luster assembly) is responsible for the general Fe-S cluster housekeeping functions, and drives the biogenesis and transfer of Fe-S clusters to the target apoproteins in *E. coli* [8,10]. The *isc* operon (iscRSUA-hscBA-fdx-iscX) is controlled by the autoregulatory transcriptional repressor IscR (*i*ron–*s*ulphur *c*luster *r*egulator), which is an established global regulator associated with the expression of at least 40 different genes as part of the response cascades that are triggered by oxidative stress [7,10,11,12,13]. The IscR protein harbours a [2Fe–2S] Fe-S cluster that natively acts as a sensor towards intracellular ROS-inflicted damage and serves as the control circuit used in the current work. As the system is based on the intrinsic chemical reactivity of the Fe-S cluster, which can be readily destabilized upon loss of electrons, it is sensitive to different types of oxidizing agents via nonspecific redox interactions. Under non-oxidizing conditions, the IscR Fe-S cluster is intact and enables the protein to bind repressor sites on specific target promoters, including P_iscR_, to block transcription. Upon oxidative stress, the Fe-S cluster is damaged, which leads to IscR dissociation from the target promoter, transcriptional derepression [6], and induction of the corresponding downstream reactions to protect the cell. Besides the Fe-S cluster maintenance, downstream protective mechanisms ultimately triggered by cellular oxidative stress include the expression of catalase hydroperoxidase I (catalase HPI), Mn- and Fe-dependent superoxide dismutases (Mn-SOD and Fe-SOD), and reduced glutathione (GSH) that scavenge oxidizing agents, in addition to various enzymes involved in DNA repair.

Understanding the mechanisms that result in microbial oxidative stress responses is vital in various areas of biological research and analytics. The progression of the ROS response provides information on the cellular response network, including the hierarchy of the components involved in conveying the ROS-induced signal, or the interplay of downstream effectors that neutralize the reactive chemical species and help the cell recover from the induced damage. Ability to measure the onset of the ROS response, which typically indicates exposure to unfavourable conditions, could help us identify immediate stress factors, such as presence of toxins or adverse culture conditions, for example, as part of bioprocess optimization. In some cases, it may also be important to distinguish between a primary physiological change (e.g., physical cellular damage) from subsequent ROS-mediated secondary effects to understand the sequence of complex biological events. However, direct quantitative detection of ROS in biological samples is complicated due to their reactive nature and consequent instability [14,15]. The most commonly used analytical methods are based on the use of chemical probes that penetrate target cells and react with different ROS species to form stable compounds that can be detected by fluorescence, chemiluminescence, and electro-paramagnetic resonance (EPR) [16,17,18]. Many of these probes are reduced dyes, which are oxidised in the presence of ROS in vivo to produce a corresponding signal that can be directly measured. These probes can also be easily oxidised by other factors, such as light [19,20,21,22], peroxynitrite, tyrosine, or thiols [23], which complicates analysis by increasing the background signal [24]. The probes or the resulting derivatives may also undergo other unwanted reactions, such as degradation by ROS scavengers, which may effectively reduce the sensitivity of the system resulting in false negative signals [18,19,25]. In addition, the diffusion of the probes into target cells can also pose a barrier and directly limit the quantitation of ROS in vivo [26]. Consequently, alternative strategies for ROS detection that provide higher selectivity or sensitivity and reduced cost would be advantageous for some applications.

This study aimed at developing an enzyme-based biological system for measuring oxidative damage in bacterial cells as a complementary strategy to the existing methods using chemical probes for in vivo ROS analysis. The idea was to exploit the native oxygen-sensitive iron–sulphur cluster protein IscR as a negative regulator for fluorescent marker protein expression to generate a plasmid that works as an oxidative stress indicator in *E. coli* [6,11,12]. The objective was then to monitor the sensitivity of the reporter system towards supplementation of hydrogen peroxide and changes in cultivation temperature and compare it with a commonly used commercial chemical fluorescence probe for ROS.

## 2. Materials and Methods

### 2.1. Microbial Strains and Standard Cultivation Conditions

*E. coli* strains DH5α, BW25113, and BW25113 ∆*ytfE* were used as the expression hosts in the study (Table 1). The cells were cultivated at +37 °C in Luria-Bertani (LB) (Catalogue number: L3022–1 KG, Missouri, Sigma-Aldrich, USA) liquid medium via shaking (200 rpm) or on plates containing 1.5% agar (Catalogue number: A1296–500 g, Sigma-Aldrich, St. Louis, MO, USA). Ampicillin (100 μg·mL^−1^) was added to maintain plasmid selection pressure in transformed strains. To induce expression, IPTG (Catalogue number: RO392, Thermo Scientific, Waltham, MA, USA) 0.5 mmol L^−1^ was supplemented in the cultures.

### 2.2. Assembly of the Reporter Plasmids and the Corresponding Expression Strains

The expression system for monitoring oxidative damage in *E. coli* was composed of two core elements, the reporter [i.e., E2-Crimson [27] fluorescence marker gene fused with a destabilising PEST sequence [28] under the control of redox-responsive P*_iscR_* from *E. coli* str. K-12 substr. MG1655], and the repressor [i.e., *iscR* gene from *E. coli* str. K-12 substr. MG1655 [6] under the control of IPTG-inducible promoter P_T5_], as described in Figure 1 and Appendix A. The sequence shown in Appendix A was synthesised commercially (DNA 2.0, Newark, CA, USA) and subcloned as a EcoRI-SacI fragment into plasmid pJ404 to generate the final reporter vector pJ404-IscR-C (Figure 1). The parallel control vector pJ404-C that does not express IscR was modified from pJ404-IscR-C by removing the *iscR* gene as an EcoRI fragment. The plasmids (pJ404-IscR-C, pJ404-C, and pJ404-empty) were transformed [29] in *E. coli* DH5α, BW25113, and BW25113 ∆*ytf**E*. After selection on ampicillin-containing plates, the resulting five transformants (DH5α/pJ404-IscR-C, DH5α/pJ404-C, DH5α/pJ404-empty, BW25113/pJ404-IscR-C, BW25113 ∆*ytfE*/pJ404-IscR-C) (Table 1) were subjected to oxidative stress analysis.

### 2.3. Oxidative Stress Analysis of the Reporter Strains Using a Microplate Reader

The response of the strains DH5α/J404-IscR-C, BW25113/pJ404-IscR-C, BW25113 ∆*ytfE*/pJ404-IscR-C, DH5α/pJ404-C, and DH5α/pJ404-empty to oxidative stress was evaluated based on fluorescence and cell growth. Pre-cultures were cultivated overnight under the default conditions and used for inoculating three main parallel cultures, which were incubated in the presence of ampicillin (100 μg·mL^−1^) and IPTG 0.5 mmol L^−1^ for about 2.5 h to reach optical density at 600 nm (OD_600nm_) of ~0.5. The cultures were then subjected to fluorescence analysis (E2-Crimson Ex_611nm_/Em_646nm_) and growth measurements (OD_600nm_) on 96-well plates using Infinite M200 PRO microplate reader (Tecan, Switzerland). After this initial reference measurement (timepoint 0), the cultures were divided into parallel batches representing the oxidative stress samples and the corresponding controls. Oxidative stress was induced by supplementing the cultures with H_2_O_2_ immediately after the timepoint 0, and subsequently at one-hour intervals throughout the cultivation. Cold stress was induced by transferring the cultures to +23 °C after the initial incubation phase at 37 °C. The fluorescence and OD_600nm_ values were subsequently measured as before once every hour.

### 2.4. Sample Preparation for CellROX Comparison

For fluorescence microscopy comparison against CellROX Green probe (Invitrogen, USA, Catalogue number: C10444), the DH5α/pJ404-IscR-C cells grown as above were divided into three parallel cultures and incubated for an additional 2 h at (i) 37 °C (control), (ii) 23 °C (cold stress), and (iii) 37 °C in the presence of 2 mmol L^−1^ H_2_O_2_ supplied at one-hour intervals. The cultures were then supplemented with 5 μM of the CellROX Green, followed by an additional 30 min incubation in the dark at 37 °C. All samples were then subjected to simultaneous analysis of red fluorescence (E2-Crimson expression from pJ404-IscR-C) and green fluorescence (CellROX Green) using confocal fluorescence microscopy.

### 2.5. Image Acquisition Using Confocal Fluorescence Microscopy for ROS Analysis

The samples were transferred to Nunc Lab-Tek 8-chambered cover glasses, and 2D confocal images were acquired using a Zeiss AxioObserver Z1 (objective 63×, 1.4) fluorescence microscope equipped with LSM 510. All settings, such as laser intensities, detector settings, relative focal plane, and digital acquisition settings, were maintained constant, pixel density was set according to the Nyquist theorem, and the two fluorescence channels were acquired using multi-tracking to minimise crosstalk. The imaging (16 images per condition) was carried out within 15 min to avoid the loss of condition-specific effects.

### 2.6. Image Analysis and Quantification of ROS-Mediated Effects

The analysis was performed with BioImageXD, an open-source software [30]. Firstly, the green channel images were smoothened with median filtering (kernel 10-10-1) and then binarised with dynamic thresholding (kernel 20-20-1, statistics type mean, the threshold over statistic 2). Connected component labelling was then used to divide the binary image foreground into different objects (removing objects smaller than 15 pixels), and the labelled objects were quantified for the average intensity of both the green and the red channels. The analysis was confirmed by repeating it with different thresholds over statistics and average intensity calculation methods, and the results were further confirmed with standard pixel-based colocalisation analysis (Pearson correlation and Manders colocalisation coefficients). A two-tailed *t*-test for unequal sample sizes and unequal variances was used, alpha level 0.05.

## 3. Results

### 3.1. Design of a Fluorescent Sensor System for Measuring Oxidative Stress Based on IscR

A stress-responsive fluorescent expression system was designed for direct, non-invasive monitoring of oxidative stress in *E. coli* cell cultures. The system relies on the function of IscR, an oxygen-sensitive auto-regulatory transcriptional repressor, and a corresponding responsive target promoter P*_iscR_* [6], which regulates the expression of the fluorescent marker protein E2-Crimson (Figure 1). IscR contains a labile [2Fe-2S] iron–sulphur cluster that natively serves as the response switch reacting to the redox status of the *E. coli* cell [31,32]. Upon elevated oxidative stress, the Fe-S cluster of IscR is damaged, which prevents the protein from maintaining the active quaternary conformation required for binding at the P*_iscR_* repressor site, resulting in *isc* activation [31,32,33]. When a normal redox state is restored, the IscR Fe-S cluster is reassembled, allowing the protein to bind the promoter to re-repress expression [6]. In our plasmid-based sensor system pJ404-IscR-C (Figure 1 and Appendix A), the promoter P*_iscR_* was recruited to control the expression of far-red fluorescent protein E2-Crimson, therefore allowing us to monitor the intracellular oxidative stress levels directly by measuring culture fluorescence. The E2-Crimson reporter was further fused to a C-terminal 40 aa PEST tag [28] (Appendix A) in order to destabilise the protein [34,35] and to facilitate the re-establishment of the baseline fluorescence after the oxidative stress is relieved. To ensure sufficient availability of IscR for maintaining tight repression and low background fluorescence under non-oxidising conditions, the pJ404-IscR-C reporter plasmid also carries a copy of the *iscR* gene under the control of an IPTG-inducible T5 promoter [36]. In parallel, we also generated a reference construct pJ404-C without the additional *iscR* to evaluate the capacity of the endogenous IscR to repress E2-Crimson in the recombinant system.

To study the fluorescence response in different cellular contexts, the reporter plasmids were transformed in *E. coli* strain DH5α [37], used commonly in routine cloning applications, as well as in BW25113 and a derived redox-sensitive deletion mutant BW25113 ∆*ytfE* impaired in native iron–sulphur cluster repair [38,39]. The resulting final strains, including the appropriate controls, were selected based on antibiotic resistance and stored for analysis (see Table 1 for the strains generated in this study).

### 3.2. Dynamic Fluorescence Stress Response Induced by Supplemented H_2_O_2_

The performance of the redox sensor strains (Table 1) was evaluated by measuring the fluorescent response towards supplemented H_2_O_2_, which is a commonly used strategy to induce oxidative stress in cell cultures. To first determine the overall sensitivity of the system, the *E. coli* DH5α carrying pJ404-IscR-C was exposed to increasing concentrations of H_2_O_2_ (0 mmol L^−1^, 1 mmol L^−1^, 2 mmol L^−1^, 4 mmol L^−1^, and 8 mmol L^−1^) and analysed for fluorescence (Figure 2a) and cell growth (Figure 2b,c) over a four-hour cultivation period. While the 0 mmol L^−1^ reference signal remained practically unchanged, the strain exhibited a dynamic response towards supplied hydrogen peroxide, with a clear correlation between the level of oxidative stress and the recorded fluorescence (Figure 2a). Based on the profiles, 2 mmol L^−1^ was selected as the default H_2_O_2_ concentration for the subsequent experiments due to the distinct fluorescent response coupled to minimal impact on cell propagation (Figure 2b; grey line), as compared to the higher concentrations (4 mmol L^−1^ and 8 mmol L^−1^), which clearly suppressed growth.

To investigate whether the plasmid-based IscR overexpression is required for effective repression of the E2-Crimson reporter (i.e., for preventing P*_iscR_* leakage and unwanted background fluorescence), the control strain DH5α/pJ404-C regulated solely by the endogenous IscR was analysed alongside DH5α/pJ404-IscR-C (see Table 1). The results showed that the uninduced zero fluorescence measured in the absence of H_2_O_2_ in DH5α/pJ404-C (Figure 3a; blue solid line) was significantly lower than the corresponding induced signal (2 mmol L^−1^ H_2_O_2_) (Figure 3a; blue dashed line), indicating that the reporter transcription is blocked in the DH5α background even without additional IscR overexpression. This was further supported by the similar background fluorescence levels observed in DH5α/pJ404-C (Figure 3a; blue solid line) and DH5α/pJ404-IscR-C overexpressing the recombinant IscR (Figure 3a; orange solid line). Notably, the maximum fluorescence levels recorded for DH5α/pJ404-C after H_2_O_2_ supplementation (Figure 3a; blue dashed line) were considerably higher than in DH5α/pJ404-IscR-C (Figure 3a; orange dashed line), demonstrating that increased levels of IscR may attenuate the response towards oxidative stress.

Analysis of the parallel reporter strain BW25113/pJ404-IscR-C (see Table 1) showed that the overall fluorescence response to 2 mmol L^−1^ H_2_O_2_ was comparable to DH5α/pJ404-IscR-C, and again, without any unspecific background (Figure 3b vs. Figure 3a; orange lines). The growth of the BW25113 strain, however, was slower (Appendix A), and the total fluorescence levels somewhat lower than measured in DH5α (Appendix A; orange lines). In striking contrast, the corresponding fluorescence levels observed for the derived redox-sensitive mutant BW25113 ∆*ytfE*/pJ404-IscR-C (Figure 3b, Appendix A; blue lines) were substantially higher than in either DH5α (Figure 3a; orange lines) or BW25113 (Figure 3b; orange lines), both with and without supplemented H_2_O_2_.

Throughout all the experiments, the fluorescence signals remained elevated after the initial stress-induced response, instead of effectively returning to the original baseline. This observation was consistent with the continued presence of active E2-Crimson in the cells, despite the C-terminal PEST sequence introduced to destabilise the protein.

### 3.3. Lowered Cultivation Temperature Triggers an IscR-Mediated Fluorescent Stress Response

Several earlier studies have demonstrated a link between lowered temperatures and intracellular ROS formation in *E. coli* [40,41,42], a phenomenon that was initially also evidenced here as a distinct colour change towards red in culture plates stored at 4 °C (Appendix A). In order to further evaluate these effects under reduced temperatures, the DH5α/pJ404-IscR-C and BW25113/pJ404-IscR-C reporter strains were grown under 37 °C and 23 °C after the initial incubation phase at 37 °C, followed by quantitative fluorescence comparison. Unlike in the response towards H_2_O_2_ (Figure 3), notable changes were not observed over the first hour after dividing the 37 °C pre-cultures into the two alternative cultivation temperatures (Figure 4 and Appendix A). After this, however, the fluorescence signals measured at 23 °C increased steadily over the recorded time period, while the 37 °C controls remained at or near the background level. This effect could also be visualised directly from the culture flasks as an appearance of an intense red colour under lowered growth temperature after 18 h, while the 37 °C control culture remained unchanged (Figure 4c). The fluorescent profiles measured under the two temperatures were similar between DH5α (Figure 4a; orange line) and BW25113 (Figure 4b; orange line), although the response towards the temperature downshift was slower in the BW25113 strain. Again, the absence of iron–sulphur cluster repair protein YtfE clearly increased the sensitivity of BW25113 towards lowered growth temperature, as seen in the distinctively higher fluorescence levels recorded in the ∆*ytfE* background (Figure 4b; dashed blue line). Notably, towards the end of the cultivation period, the amplitude of the sustained temperature-induced fluorescent response (Figure 4) was consistently higher than the response triggered by the addition of H_2_O_2_ (Figure 3).

### 3.4. Comparison of the Reporter System with a Commercial Probe Sensitive towards Oxidising Agents

To further evaluate the acquired response in reference to established analytical methods, the plasmid-based system (DH5α/pJ404-IscR-C) was studied in parallel to a commercial chemical probe (CellROX Green fluorescent probe) commonly used in ROS detection in vivo [43]. This particular probe was selected due to its general sensitivity towards different oxidising agents, in line with the nonspecific nature of the IscR- response. For the assay, after the initial incubation at 37 °C, cells harbouring the plasmid pJ404-IscR-C were grown at the two alternative temperatures (23 °C and 37 °C) and in the presence of H_2_O_2_ at 37 °C, followed by treatment with CellROX. Subsequent confocal microscopy imaging (Figure 5 and Appendix A) allowed us to distinguish the concurrent green fluorescence signal (specific for CellROX) (Figure 5b and Appendix A) and red fluorescence signal (specific for E2-Crimson) (Figure 5c and Appendix A) in all the samples and compare them against one another. Importantly, the red and green fluorescence clearly co-localised in individual cells (Figure 6), supporting ROS as the common origin of the signals. Further quantitative analysis of the image data showed that the two reporter systems responded to the treatments similarly (Figure 7). In both cases, the supplementation of H_2_O_2_ and the reduction of cultivation temperature resulted in a statistically significant fluorescent signal increase in reference to the 37 °C control, as seen in the green channel (Figure 7a) and in the red channel (Figure 7b). In the designed plasmid-based system, the differences between the treated and the untreated samples were more pronounced, implicating a possible higher sensitivity towards changes in the ROS-induced damage (Figure 7b vs. Figure 7a). Throughout the confocal microscopy image data, the acquired red fluorescent signal also appeared to have a more uniform intensity and distribution across the cell population in comparison to CellROX, as represented in Figure 5c vs. Figure 5b.

## 4. Discussion

The ability to measure intracellular ROS is of specific importance for understanding different stress-related conditions and the resulting redox reactions in the cell. However, distinguishing between the primary oxidative effect and the metabolic response may be complicated as (i) ROS formation often induces multiple direct and indirect changes, (ii) the fluorescent dyes used for the analysis are not necessarily strictly specific to ROS [19], and (iii) the outcome may also be affected by other variables such as penetration of the probe into the cell [15]. In order to find new alternative strategies for monitoring ROS in *E. coli*, we designed a simple plasmid-based reporter system (Figure 1), which reacts to the oxidising environment inside the cell by expressing far-red fluorescent protein E2-Crimson as a quantitative marker. The control circuit is based on the oxygen-sensitive transcriptional regulator IscR, and the corresponding target promoter P*_iscR_* from *E. coli*, which natively detects oxidative damage and triggers specific response cascades to protect the cell against ROS-mediated effects [44]. The devised expression system therefore exploits a natural biological mechanism for sensing redox changes and provides a new direct and physiologically relevant approach for measuring intracellular oxidative stress.

The constructed fluorescent reporter system (Figure 1a,b, Table 1) responded to H_2_O_2_ supplementation in a concentration-dependent manner and without any considerable background (Figure 2), as measured from cultures of *E. coli* DH5α (Figure 3a) and BW25113 (Figure 3b) carrying the reporter plasmid pJ404-IscR-C. This confirmed that the reporter functioned as anticipated and resulted in E2-Crimson expression from P*_iscR_* upon ROS-induced IscR inactivation. Analysis of the corresponding system without IscR overexpression (Figure 1c and Figure 3a; green line) revealed that P*_iscR_* repression could be maintained by the endogenous IscR encoded by the *E. coli* host. This, however, does not exclude the possibility that the expression of the recombinant IscR would be necessary under some conditions for maintaining unspecific background fluorescence at minimum. At the same time, the overexpression of IscR in DH5α/pJ404-IscR-C was observed to reduce the maximum fluorescence levels in the presence of H_2_O_2_ (Figure 3a; dashed orange and dashed green line), which may have important implications in reference to the host’s own response system, as excess IscR may attenuate the native downstream effects triggered by oxidative stress. The reporter construct was also evaluated in the *E. coli* BW25113 ∆*ytfE* mutant strain, which is highly sensitive toward ROS due to impaired Fe-S cluster repair [38,39] (Figure 3b). The fluorescence levels measured for the ∆*ytfE* mutant (Figure 3b; blue line) were consistently higher than in the parent strain background (Figure 3b; orange line) or DH5α (Figure 3a; orange line), both in the absence and in the presence of H_2_O_2_. This is likely to be a consequence of increased oxidative stress in the ∆*ytfE* strain, in which the normal maintenance of cellular Fe-S proteins is compromised, thereby altering the natural response to elevated levels of H_2_O_2_. Together, these results suggest that the reporter pJ404-IscR-C can be applied to obtain quantitative analytical data on oxidative damage in intact *E. coli* cells exposed to either external stimuli (e.g., supplemented H_2_O_2_ or temperature shift) or genome-level changes that affect the sensitivity towards ROS (e.g., impaired Fe-S cluster repair in ∆*ytfE*).

In an optimal scenario, the response signal should be induced under elevated oxidative stress and attenuate as the normal intracellular conditions are re-established. In the case of the IscR/E2-Crimson system, the fluorescence is effectively triggered by H_2_O_2_ and temperature shifts, but the resulting signal does not readily return to the baseline. The fluorescence levels in the setup are concomitantly determined by (i) the efficiency of E2-Crimson expression, which is strictly regulated by the IscR switch, and (ii) the specific rate of E2-Crimson degradation that we aimed to reinforce through the introduction of a destabilising PEST sequence at the protein C-terminus. Although the PEST system has been previously mainly used in eukaryotes [28], induced protein degradation reported in *E. coli* [34] suggested that the peptide sequence should also function in prokaryotes. In our system, however, E2-Crimson destabilisation was not as effective as anticipated, which encourages other protein degradation tags, as discussed in [35,45,46], to be tested in future applications.

Comparison of the E2-Crimson reporter pJ404-IscR-C with a commercial green fluorescent redox probe CellROX [43] showed highly similar responses to oxidative stress (Figure 7). The systems were triggered by the same conditions (temperature shift and supplemented H_2_O_2_) (Figure 7) and produced equivalent signals that co-localised in the same cells (Figure 6), indicating that the two strategies could complement one another in measuring ROS in *E. coli*. With the notion that the sensitivity and resolution of the plasmid system appeared proportional or even higher in reference to CellROX under the conditions tested, one of the key advantages of pJ404-IscR-C is that it specifically reacts to biologically relevant oxidative stress levels, as the circuit relies on the damage inflicted on the native ROS-sensor IscR. The findings, therefore, suggest that this system can provide a prominent alternative to fluorescent chemical dyes in the analysis of ROS-mediated effects in *E. coli*, especially when investigating whether or not a specific stress factor would naturally initiate a ROS-response in the cell.

Suboptimal growth temperatures have earlier been shown to trigger ROS-related effects in *E. coli* [40,41,42], which was also of our interest when evaluating the performance of the designed reporter system towards oxidative damage. Accordingly, cultivation of the cells at the reduced temperature of 23 °C (Figure 4; dashed lines) resulted in a response similar to the H_2_O_2_ treatment. The temperature effect was seen in both DH5α (Figure 4a) and BW25113 (Figure 4b) harbouring the reporter plasmid pJ404-IscR-C, as well in the CellROX analysis used for comparison (Figure 7). This supported the view that lowered temperature exposes cells to increased intracellular oxidative stress that triggers Fe-S cluster damage, resulting in the inactivation of IscR and subsequent expression of the E2-Crimson in the reporter strains. The cold stress effect has been previously discussed in context with indirect transcriptomics data acquired under different stress conditions, which revealed clear associations between the oxidative response and the temperature decrease at the mRNA level [47]. This has also been investigated directly in redox-sensitive *E. coli* mutant strains, demonstrating that the metabolic effect is fundamentally similar to the oxidative stress response triggered by H_2_O_2_. However, based on the mutant strains that specifically lack superoxide dismutase (Mn-SOD, Fe-SOD) activity, it appears likely that the effect is linked to increased accumulation of superoxide anions rather than H_2_O_2_ [40]. The effect has been proposed to be caused by lowered respiration and consequent relative increase of dissolved intracellular oxygen at lower temperatures, which, based on our findings, is also clearly detectable in the normal DH5α and BW25113 background.

The data presented here show that the developed sensor system is applicable for analysing oxidative damage in *E. coli* under varying conditions. Supported by the existing knowledge on IscR, the findings reinforce the view that the response is rather general towards any oxidising agents that react with and destabilize the repressor Fe-S cluster, rather than being specific towards particular forms of ROS. This makes the system well-suited for monitoring the overall global oxidative stress status of the cell, and the biological effects that are natively conveyed by IscR. As implied by the comparison in DHα and BW25113, the reporter plasmid is expected to function in different *E. coli* strains that are compatible with the pJ404 backbone vector. Requiring only the transformation of the target strain with pJ404-IscR-C, the method is fast and inexpensive compared to commercial chemical dyes, which must always be separately supplied to each measured sample. The plasmid-based approach can, therefore, provide a practical alternative in experimental setups where the supplementation of redox chemicals in the cell sample is restricted, or when substrate penetration and diffusion limit the response time or the signal distribution. The reporter cassette can also be subcloned as a separate module into some other *E. coli* plasmid to study the ROS-related effects in a specific microbial expression system, while the sensitivity could further be optimised by adjusting the expression level of IscR to suit the particular purpose. Notably, the use of the sensor in other microbes than *E. coli* may require modifications in the repressor circuit or the fluorescent reporter to match the redox environment or other species-specific features of the host. In *E. coli,* the devised system can be directly applied to investigate general oxidative effects of chemical supplements (e.g., antibiotics or antioxidants), growth conditions (e.g., heat, pH, or osmotic stress), or genetic modifications that can indirectly result in oxidate stress. By design, the reporter is expected to be well-applicable for studying environmental cues such as iron starvation or nitrosative stress, which readily induce Fe-S cluster damage and inactivation the corresponding enzyme functions.

## Figures and Tables

**Figure 1 sensors-22-06334-f001:**
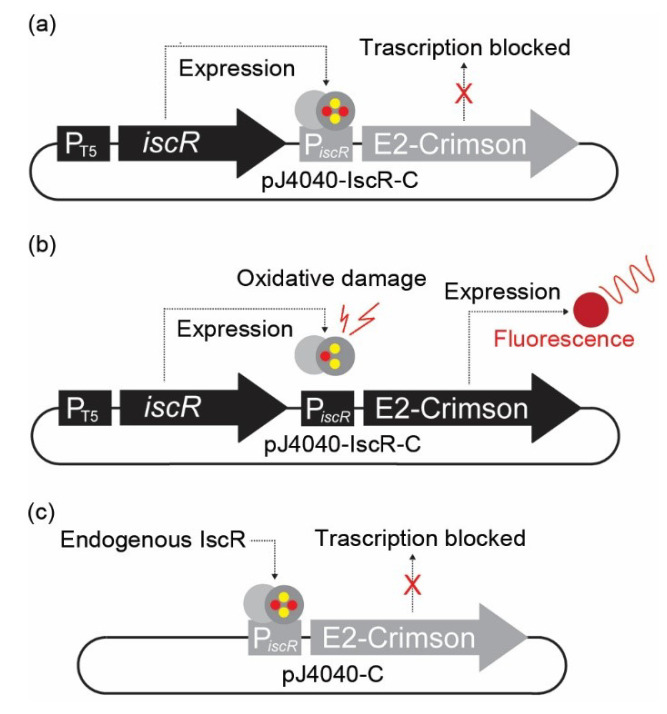
**Representation of the plasmid-based in vivo fluorescent reporter system for Fe-S cluster damage designed in this study.** (**a**) In pJ404-IscR-C, the E2-Crimson gene expressing the fluorescent reporter protein has been placed under the regulation of *iscR* promoter (P*_iscR_*), which is effectively suppressed in the presence of intact transcriptional repressor IscR. Consequently, no fluorescent signal is detected under normal conditions when IscR is expressed under the regulation of IPTG-inducible T5 promoter. (**b**) Under oxidative stress, the IscR [2Fe-2S] cluster is damaged, and the protein can no longer bind the promoter repressor region. This results in the expression of E2-Crimson protein and allows the detection of a red fluorescence signal. (**c**) In the modified construct pJ404-C, the IscR protein is not overexpressed from the reporter plasmid, and the repression of E2-Crimson relies on the endogenous IscR present in the *E. coli* host.

**Figure 2 sensors-22-06334-f002:**
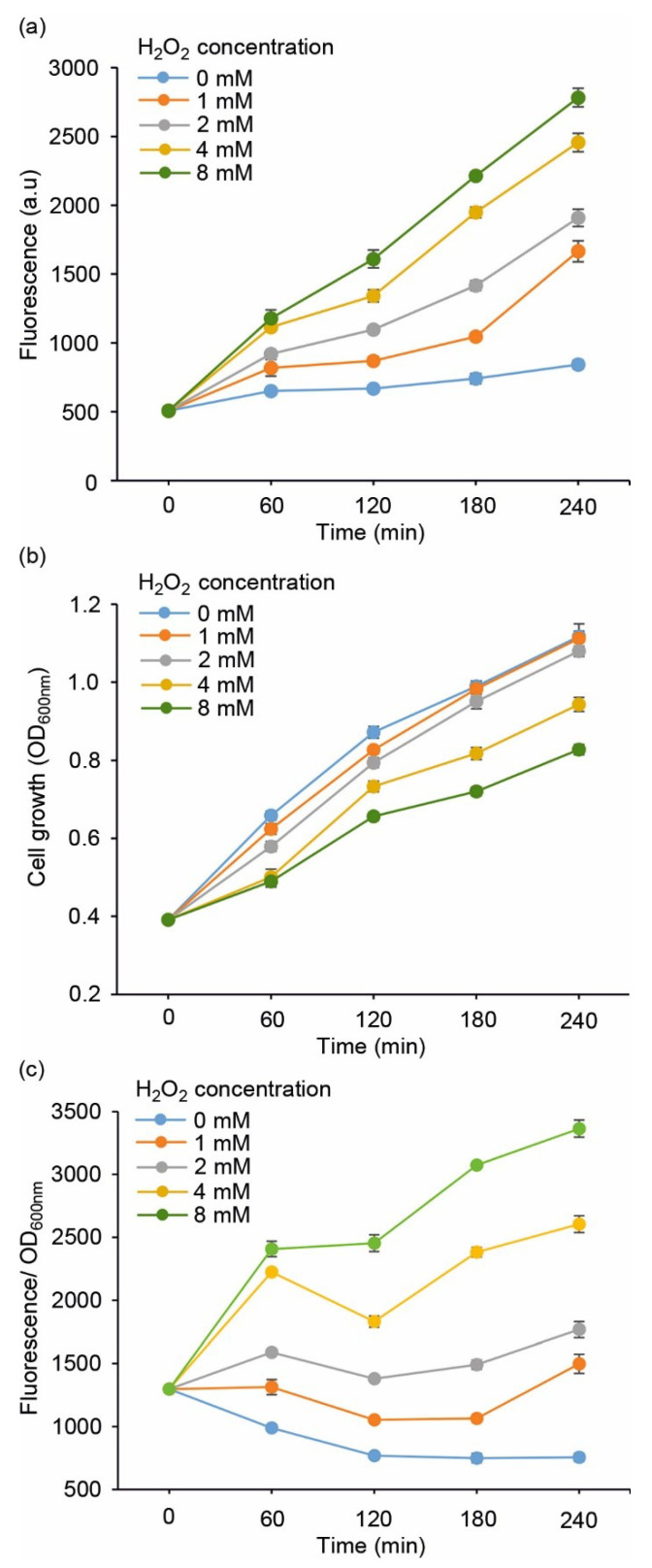
**Evaluating the performance of *E. coli* DH5α/pJ404-IscR-C sensor system towards varying concentrations of supplemented H_2_O_2_**, monitored as (**a**) far-red E2-Crimson fluorescence, (**b**) cell growth (OD_600nm_), and (**c**) fluorescence normalised for growth at OD_600nm_. The values were measured at one-hour intervals from cell cultures repeatedly supplemented with H_2_O_2_ corresponding to 1 mmol L^−1^ (orange line), 2 mmol L^−1^ (grey line), 4 mmol L^−1^ (yellow line), and 8 mmol L^−1^ (green line) every hour, in reference to cultures without H_2_O_2_ (light blue line). The averages and standard deviations were calculated based on three independent parallel cultures with three technical replicates each (*n* = 9). Due to small standard deviations, the error bars are not always visible.

**Figure 3 sensors-22-06334-f003:**
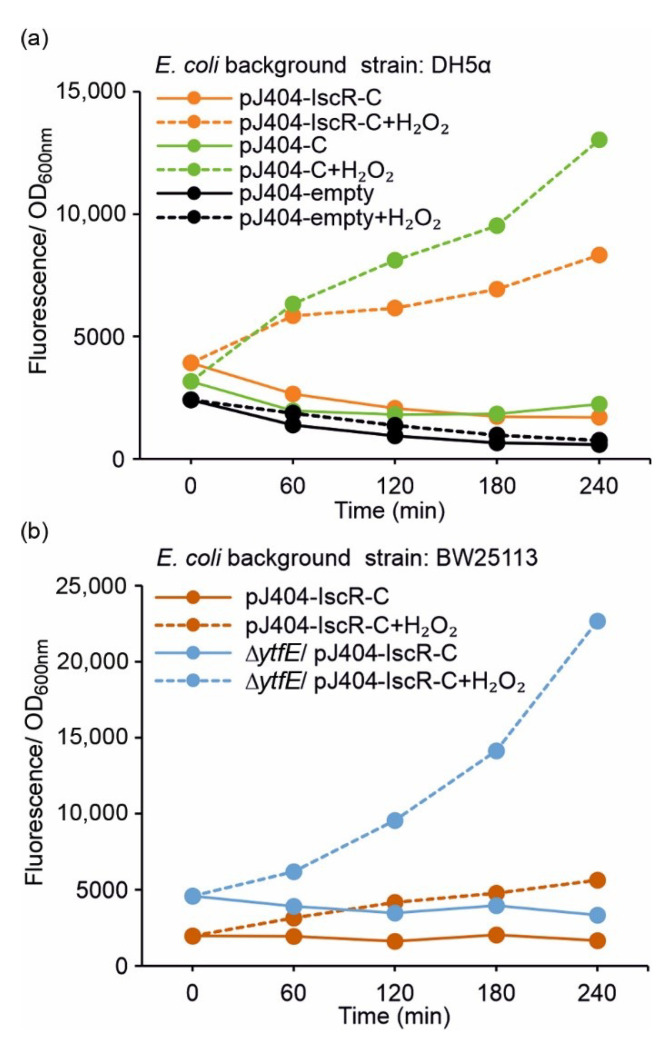
**Fluorescence response of the alternative redox sensor systems towards oxidative stress induced by 2 mmol L^−1^ supplemented H_2_O_2_** in (**a**) DH5α and (**b**) BW25113 (/∆*ytfE*) background (see Table 1 for strain details). The fluorescence values were measured at one-hour intervals from cultures repeatedly supplemented with 2 mmol L^−1^ H_2_O_2_ (dashed lines) and cultures without H_2_O_2_ addition (solid lines)_._ The averages and standard deviations were calculated based on three independent parallel cultures with three technical replicates each (*n* = 9). Due to small standard deviations, the error bars are not always visible.

**Figure 4 sensors-22-06334-f004:**
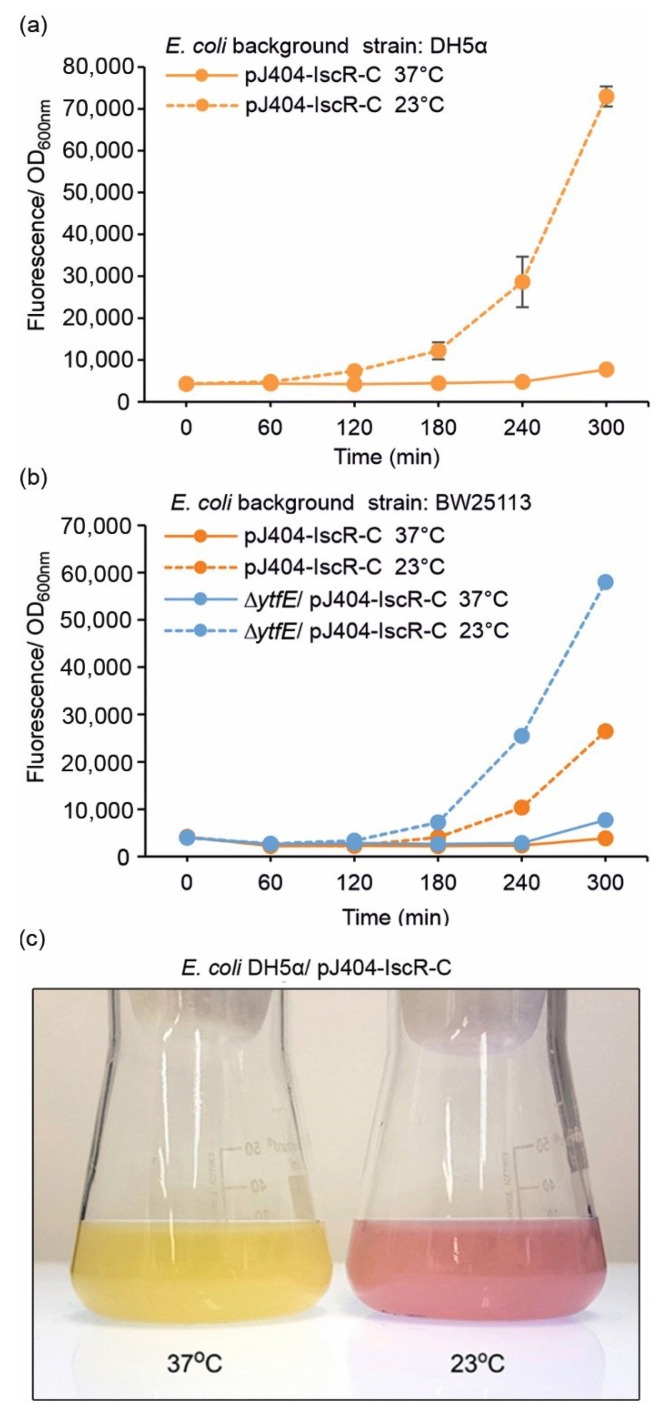
**Fluorescence response of the pJ404-IscR-C sensor system towards lowered cultivation temperature in** (**a**) DH5α and (**b**) BW25113 background (see Table 1 for strain details). Fluorescence was measured at one-hour intervals from cultures grown at the default temperature of 37 °C (solid lines) or 23 °C (dashed line). The averages and standard deviations were calculated based on three independent parallel cultures with three technical replicates each (*n* = 9). Due to small standard deviations, the error bars are not always visible. (**c**) Visualisation of the red fluorescence response triggered by lowered cultivation temperature, as observed directly from parallel flask cultures of the generated *E. coli* reporter strain DH5α/pJ404-IscR-C incubated at 37 °C (left) and 23 °C (right) for 18 h after induction.

**Figure 5 sensors-22-06334-f005:**
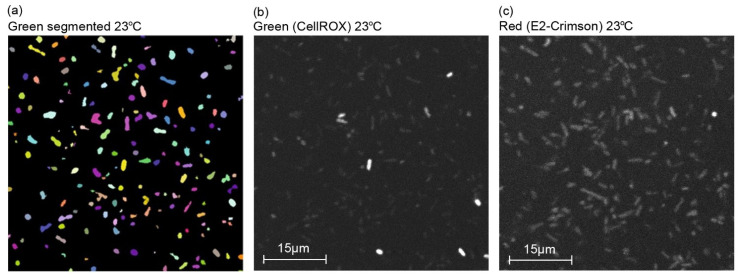
**Representative confocal microscopy images of *E. coli* reporter strain DH5α/pJ404-IscR-C grown at 23 °C subjected to oxidative stress analysis**. (**a**) Green fluorescence channel after segmentation with dynamic thresholding to separate bacteria from the background, (**b**) original image of the cells emitting green fluorescence corresponding to the ROS-sensitive commercial probe CellROX, and (**c**) original image of the cells emitting red fluorescence corresponding to the IscR-regulated E2-Crimson reporter system.

**Figure 6 sensors-22-06334-f006:**
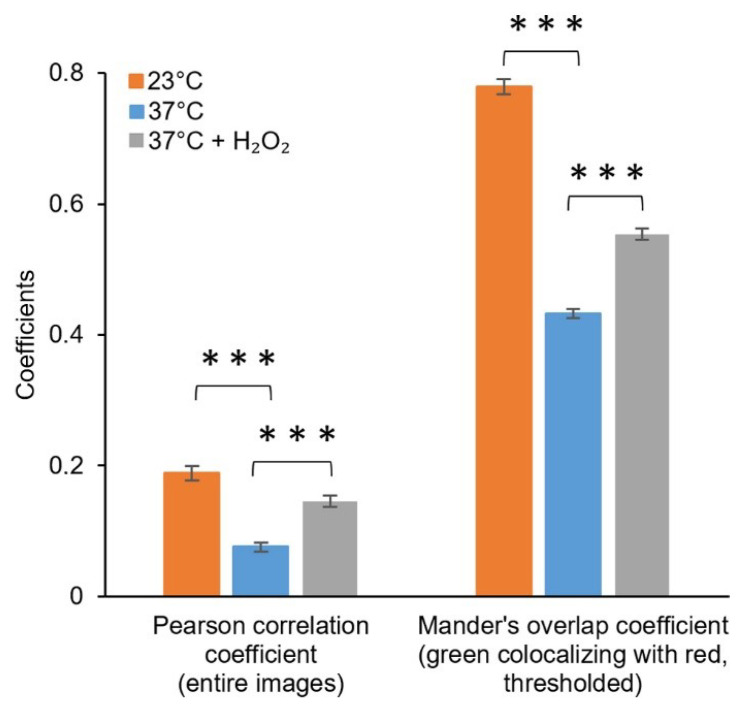
**Colocalisation of the fluorescence signals from the green channel (CellROX) and red channel (pJ404-IscR-C/E2-Crimson) acquired under control conditions and oxidative stress-inducing conditions.** Fluorescence intensities from the entire confocal microscopy images were used for calculating the Pearson correlation coefficient, while the Manders’ colocalisation coefficients (only M1 shown) were calculated using threshold images. The differences between the samples are statistically significant, with *p* ≤ 0.001 (***).

**Figure 7 sensors-22-06334-f007:**
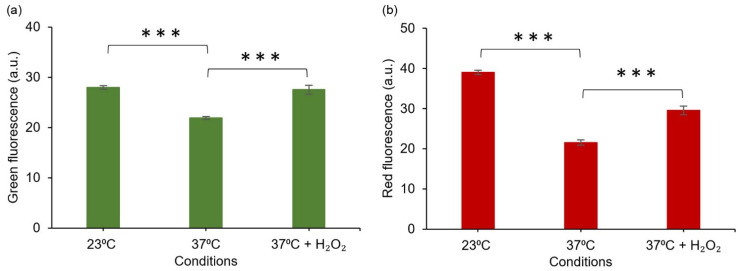
**Fluorescent response of *E. coli* DH5α/pJ4040-IscR-C towards intracellular oxidative stress induced by reduced temperature (23 °C) and H_2_O_2_ supplementation, based on quantitative analysis of confocal microscopy images.** (**a**) Green fluorescence signal representing the commercial CellROX probe and (**b**) red fluorescence representing the designed E2-Crimson reporter system, as determined from corresponding segmented *E. coli* objects. The averages and standard deviations have been calculated based on three independent datasets. The differences between the samples are statistically significant, *p* ≤ 0.001 (***).

**Table 1 sensors-22-06334-t001:** **List of the *E. coli* strains generated and analysed in this study.** “IscR” in the plasmid name indicates IscR overexpression, while “C” refers to the gene encoding the fluorescent E2-Crimson. See Figure 1 and Appendix A for plasmid construct details.

Background *E. coli* Strain	Plasmid	Description
**DH5α**	pJ404-IscR-C	The default reporter system to monitor oxidative damage
**DH5α**	pJ404-C	Reference system without IscR overexpression
**DH5α**	pJ404 empty plasmid	Negative background control (fluorescence baseline)
**BW25113**	pJ404-IscR-C	Reporter system in alternative host and Reference for BW25113 ∆*ytfE*
**BW25113 ∆ytfE**	pJ404-IscR-C	Reporter system in redox-sensitive host strain

## Data Availability

All data are provided in the manuscript and Appendix A.

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
