# Peer review of "A Plasmid-Based Fluorescence Reporter System for Monitoring Oxidative Damage in E. coli"

_sensors, 2022, doi:10.3390/s22176334_

Round 1
Reviewer 1 Report
1. Please include Isopropyl Β-D-1-thiogalactopyranoside (IPTG) in the abbreviations list.
2. Lines 154-160 should be omitted, since they belong to the journal instructions.
3. In the M&M section, you should provide detailed information about the CellROX experiments.
4. Your results could be more convincing if Figure 5 was expanded to include confocal microscopy images from all experimental variations (e.g. plasmids, bacterial strains, temperatures).
5. In the Discussion section, please provide some addition information regarding the specificity of your assay. Which are the possible interferants? How broad is the group of compounds involved in the oxidation of the FeS moieties? Which are the possible interpretations of a positive score using your assay as far as the type of affected metabolic pathways is concerned?
Reviewer 2 Report
In this work, Dandapani and co-workers proposed a novel fluorescent system for oxidative stress detection in bacterial cells. This is based on a plasmid-based system that relies on an oxygen-sensitive transcriptional regulator IscR to control the expression of a fluorescent marker in E. coli. This approach is interesting and original, and was able to produce fluorescence as a response to oxidative stress. The performance of this system was comparable to commercial chemical sensors. Thus, this study should be of interest for the readership of Sensors.
Nevertheless, there are some aspects that the authors must address in Major Revision. Namely:
-Figure 3 requires error bars;
-Why the choice for CellROX as a reference probe?
-To better compare their system with commercially available approaches, the authors should include more reference probes?
-Does the proposed system responds to oxidative stress in general, or to specific reactive oxygen species?
-In the introduction section, the authors should focus on why there is an need to measure oxidative stress specifically in bacterial cells;
-If the proposed system is comparable to chemical sensors, what is the advantage of the proposed system?
-In the manuscript, the authors write about quantitative oxidative stress, but not relevant demonstration was made of the ability of this system to perform quantitative analysis of ROS detection/measurement;
Round 2
Reviewer 1 Report
Well done.
Reviewer 2 Report
The authors have addressed my comments, and so, my recommendation is for acceptance.